# Microsatellite Instability and Immune Response: From Microenvironment Features to Therapeutic Actionability—Lessons from Colorectal Cancer

**DOI:** 10.3390/genes14061169

**Published:** 2023-05-27

**Authors:** Luana Greco, Federica Rubbino, Arianna Dal Buono, Luigi Laghi

**Affiliations:** 1Laboratory of Molecular Gastroenterology, IRCCS Humanitas Research Hospital, Via Manzoni 56, 20089 Rozzano, Italy; luana.greco@humanitasresearch.it (L.G.); federica.rubbino@humanitasresearch.it (F.R.); 2Division of Gastroenterology, Department of Gastroenterology, IRCCS Humanitas Research Hospital, Via Manzoni 56, 20089 Rozzano, Italy; arianna.dalbuono@humanitas.it; 3Department of Medicine and Surgery, University of Parma, 43126 Parma, Italy

**Keywords:** microsatellite instability, DNA mismatch repair, colorectal cancer, hereditary syndrome, immunotherapy, microenviroment, immune checkpoint, checkpoint inhibitors

## Abstract

Microsatellite instability (MSI) can be found in 15–20% of all colorectal cancers (CRC) and is the key feature of a defective DNA mismatch repair (MMR) system. Currently, MSI has been established as a unique and pivotal biomarker in the diagnosis, prognosis, and treatment of CRC. MSI tumors display a strong lymphocytic activation and a shift toward a tumoral microenvironment restraining metastatic potential and ensuing in a high responsiveness to immunotherapy of MSI CRC. Indeed, neoplastic cells with an MMR defect overexpress several immune checkpoint proteins, such as programmed death-1 (PD-1) and programmed death-ligand 1(PD-L1), that can be pharmacologically targeted, allowing for the revival the cytotoxic immune response toward the tumor. This review aims to illustrate the role of MSI in the tumor biology of colorectal cancer, focusing on the immune interactions with the microenvironment and their therapeutic implications.

## 1. Introduction

Colorectal cancer (CRC) is the second most common non-sex-dependent type of cancer, accounting for over 500.000 new cases and being the second most common cause of cancer death in Europe in 2020 [1]. These rates feed the need of implementing screening policies [2] altogether with the knowledge of the biological bases of its initiation and progression pathways to appropriately restrain morbidity and mortality.

CRC is a heterogeneous and multi-factorial disease; however, it is classifiable by its genetic bases: alongside the classical model of tumorigenesis (i.e., the chromosomal instability pathway, or CIN), a subset of CRC is marked by the presence of microsatellite instability (MSI), accounting for 15% of all cases (overall 3% being hereditary and the remaining 12% sporadic) [3,4]. The molecular phenotype of MSI was first recognized in the early 1990s [5]; such a phenotype originates from the presence of deletions in repetitive DNA sequences (called microsatellites) due to a dysfunction of the DNA mismatch repair system (MMR), the highly conserved system for correcting the errors that develop during DNA replication. A defective MMR (dMMR) system ensues, therefore, in constantly high mutational rates and instability of the tumor genome, leading to the accumulation of frameshift mutations and thus of truncated peptides [6,7]. MSI CRC exhibits distinct clinical features, including a more frequent location in the right colon, a better prognosis, a prominent immune infiltrate in and around the tumor tissue, and frequently a poorly differentiated, mucinous, or signet ring appearance [3,8].

As mentioned, like CIN CRCs, MSI CRCs can also have a hereditary or sporadic origin [3,4]. Whenever a germline pathogenic variant (PV) in one of the MMR genes is harbored (i.e., *MLH1*, *PMS2*, *MSH2*, *MSH6*, or of epithelial cell adhesion molecule—*EPCAM*), this determines Lynch syndrome (LS) [9,10]. Particularly, deletions involving *EPCAM*, the gene upstream of *MSH2*, are also considered causative of LS because they induce *MSH2* promoter methylation and silencing [11,12,13].

LS is the most frequent autosomal dominant condition predisposing to CRC, also carrying an increased risk of extracolonic cancers (i.e., stomach, small bowel and pancreas, endometrium and ovaries, and urinary tract) [9,10], with variable penetrance depending on the affected gene. LS presents frequently with young-onset and synchronous or metachronous tumors [14,15]. For its autosomal dominant heritability, LS must be evaluated in all relatives of affected patients to select all family members that might benefit from personalized surveillance programs. Although here we refer to LS as a uniform entity, this is not the case, as PVs in each of the four MMR genes lead to rather different clinical manifestations, which impact cancer development in the different organs [16] that may be affected, and thus their prevention as well [17]. However, most MSI CRCs have a sporadic origin due to somatic methylation of the *MLH1* promoter, which is frequently coupled with the missense mutation in exon 15 of the *BRAF* gene, resulting in a change at residue 600 that substitutes glutamine for valine (c.1799C > A, V600E) [3,5]. Universal screening of the MS status of all CRCs is recommended to identify individuals with LS among those affected by tumors displaying MSI, which as a biomarker also brings prognostic and therapeutic implications [18]. Due to their high mutational burden, MSI cancer cells express a high number of immunogenic soluble and surface neoantigens (aberrant truncated peptides), which trigger an important local and systemic [19] immune response, at first testified by the histological characteristic of a dense lymphocytic tumor infiltrate [6,7]. Recently, it has been hypothesized that such a relevant immune response may affect the development of MSI tumors in the colon, potentially achieving their clearance [20]. In any event, this important local activation of the immune system represents one of the explanations for the better prognosis and the low tendency to metastasize of MSI cancers [21], the other one possibly residing in the excessive mutational burden, which may not favor clonal selection [22]. The recognition by the adaptive immune system of tumoral neoantigens eventually generates an antitumor response able to restrain tumor proliferation [22]. However, neoplastic cells with dMMR eventually will overexpress several immune checkpoint (IC) proteins, such as PD-1 e PD-L1, which act to restrain the efficiency of the cytotoxic immune response. The pharmacological inhibition of these ICs through immunotherapy (such as Pembrolizumab and Nivolumab) allows for overcoming such blockade of the immune response, restoring an active adaptive response toward cancer cells [23,24,25].

The understanding of the peculiar carcinogenesis of MSI tumors altogether with that of the related acquired immune response has extensively grown since the discovery of dMMR CRC and exerted an enormous clinical impact in gastrointestinal oncology. In this review, we aim to summarize the peculiar features of MSI colorectal cancer and the related clinical implications for their management, with a focus on immune interactions within the tumor microenvironment and their therapeutic exploitation.

## 2. The Genomic Landscape of MMR-Deficient Cancers

After an initial discovery phase of the involvement of the MMR genes in the molecular pathogenesis of LS, two workshops held at The National Cancer Institute operatively defined MSI-high (MSI-H) as the molecular hallmark for dMMR tumors if 40% of microsatellite *loci* (out of five) showed frameshift mutations, while those with one frameshifted marker only were defined as MSI-low (MSI-L, a definition progressively abandoned) [26,27].

Irrespective of their inherited or sporadic nature, MSI-H CRCs accumulate frameshifts in repetitive DNA tracts (mostly being poly-A/T runs) within the coding sequences of driver genes such as Transforming Growth Factor β Receptor 2 (*TGFBR2*), Activin A Receptor Type 2A (*ACVR2*), *MSH3* and *MSH6*, and many others widespread in the genome. These mutations, wherever occurring, originate stop codons, eventually resulting in truncated proteins acting as neoantigens for the immune system [28], inactivating an array of pathways. Looking closely, it also emerged that among sporadic MSI, ≥40% of the cases show the *BRAFV600E* mutation (although this gene harbors no coding microsatellite) [3,29,30] altogether with *MLH1* hypermethylation and activation of WNT signaling, causing the hyperactivation of the EGFR/RAS/RAF/MAPK pathway, promoting cell proliferation [31].

Besides inherent gene alterations, MSI CRCs are associated with an increased expression of genes related to a diffuse immune infiltrate such as T-box transcription factor 21 (*TBX21*) encoding T-box expressed in T cells (TBET) protein, the Th1/Tc1 cytokine canonical transcription factor, interferon-γ (*IFNG*), CD8a molecule (*CD8A*), granzyme B (*GZMB*), perforin 1 (*PRF1*), and interleukin 21 (*IL21*) expressed by cytotoxic T lymphocytes (CTLs) [32]. A gene expression analysis from TCGA established that a low expression of MMR-deficient genes, including *MLH1*, *MLH3*, *PMS1*, and *PMS2* as well as other double-stranded break DNA that repair genes including *ATR*, *PRKDC*, *ATM*, and *BRCA2*, was associated with MSI-H [33].

Later, with the advent of next-generation sequencing, the Cancer Genome Atlas Network classified (sporadic) CRCs into hypermutated (16%) and non-hypermutated (84%) ones, with clear separation among the two groups. While hypermutated tumors are characterized by mutation rates up to 12/10 megabases and by a median number of non-silent mutations up to 700, mutation rates in non-hypermutated tumors are <8/10 megabases, with a median number of 58 non-silent mutations. Somatic hypermethylation of the *MLH1* promoter causes most hypermutated sporadic MSI CRCs, whereas somatic biallelic inactivation of MMR genes occurs in those remaining. These somatic events characterize about two thirds of the cases of “Lynch-like” patients without germline mutations of MMR genes and hypermethylation of *hMLH1* in MSI CRC (Lynch-like syndrome), with onset generally between the fourth and fifth decades of age [17,34,35,36,37,38,39,40].

### MSI, a Distinguishable Genetic Attractor of the Adaptive Immune Response

Since the recognition of MSI CRCs, their prominent infiltration by T lymphocytes has been portrayed as a morphological hallmark [41,42,43], up to the point that such a feature, referred to as a “Crohn-like reaction”, was included among the Bethesda criteria for MSI testing [27]. As a clinical correlate and possible expression of a protective function of dense tumor-infiltrating lymphocytes (TILs), it was noticed that their load at both the tumor invasive front and core positively correlated with a better outcome [44,45,46,47].

While the genomic landscape of CRC became a land of innovative discovery for molecular oncology [48], immunologists took an independent approach to unravelling the relevance of the adaptive immune response to disease outcome [46]. Such an approach, based upon the prognostic impact of the density of TILs, revived previous observations [49] and established that the amount of the adaptive immune response is one of the main determinants of disease behavior [50]. This immunological approach did not initially consider the inherent differences between MSI and MSS CRCs as to their TIL loads. However, it has progressively emerged that MSI CRCs account for a large fraction of the cases explaining the optimal outcome of CRCs with a high density of immune infiltrate, especially among early-stage ones [51]. Undisputedly, MSI CRCs with high immune infiltration as well have a better outcome than their counterpart with low infiltration [51,52,53]. Stage dependency of this evidence is still a matter of debate, as a difference likely exists between the crude prognostic value of TILs in untreated CRCs (that is, stage II) [54] and in those with nodal involvement (i.e., stage III) [55,56], which are treated according to shared standards with adjuvant therapy. In any event, the relevance of the immune landscape of CRC shaped the scenario for a different view on CRC outcome. The intersection between genetics and immunology eventually led to identifying MSI CRCs as the optimal target for immunotherapy due to the expression of multiple counter-inhibitory checkpoints in the background of a robust immune response [32]. Indeed, the number of frameshift mutations correlates with clinical responses to immunotherapy: tumors with low mutational burden are less responsive than those with higher ones [57].

Consistently, the heterogeneous immune landscape of CRCs could be largely predicted by their MS status, the exception being the tumors with *POLE* mutations [58,59,60], which develop a profile of immune response such as that observed in MSI CRCs. Thus, while MSS CRCs harbor a variable infiltration of TILs, most MSI tumors have the highest, such as the expression of neoantigens that trigger such an antitumor response [8,61]. Intra-epithelial CTLs (especially γ-Delta) and activated CD4+ helper T-cells infiltrate MSI tumors, making them progressively amenable to a local cytotoxic immune response [33,62]. In this environment, IC molecules encompassing PD-1, PD-L1, CTLA-4, LAG-3, and TIM3 act as a ‘co-inhibitory’ or ‘checkpoint’ receptor expressed on the IFN-γ-producing helper and CTLs [63,64,65,66,67] and are higher in MSI CRCs than in MSS ones [33]. Interestingly, it has also been shown that immune changes in CRC can vary depending upon specific settings, so that patients with synchronous tumors, which show dissimilar somatic alterations and different clonality, harbor damaging germline mutations in immune-related genes [68].

Maby et al. highlighted the link between MS status and the high prevalence of frameshift mutations, which in turn correlated with the load of CD3+, CD8+, and FOXP3+ TILs [69]. However, Llosa and coll. recapitulated that MSI CRCs are characterized by a higher density of memory effector cells and Th1, as well as proliferating T cells, counterbalanced by the expression of IC, such as PD-1 (programmed cell death molecule), PD-L1 (PD1 ligand), CTLA-4 (cytotoxic T-lymphocyte associated protein 4), LAG-3 (lymphocyte activation gene), and IDO (Indoleamine 2,3-Dioxygenase 1) [32], albeit they cannot be assumed as indicators of clinical response [70,71], as well as cytotoxic mutation-specific T cells [72].

The enrichment of inflammatory IL-17 and TNF-α Secreting CD4(+) T cells was observed in MSS CRC specimens compared with adjacent uninvolved tissue, as well as of PD-1. Consistently, the frequencies of CD25 + CD127^low^ Treg cells seem to be inversely related to total T cell IFN-γ and IL-2, suggesting a possible inhibition of antitumor effector responses [73].

A comparison of immunological characteristics between paired dMMR and MMR-proficient (pMMR) CRCs performed by Liu et al. showed that MMR status significantly correlated with the expression of MHC class I and CD8+ cells, with dMMR group displaying significantly less MHC class I but higher CD8 expression than the pMMR group (all *p* < 0.01). Consistently, low MHC class I and CD4 expression and high CD8 expression were significant predictors of dMMR (odds ratio (OR) = 24.66, 2.94, and 2.97, respectively; all *p* < 0.05), dMMR status being the only significant predictor for MHC class I low expression (OR = 15.34; *p* < 0.001) [74].

A recent systematic review evaluated distinct immune phenotypes in hereditary and sporadic MSI CRCs, distinguishing immune infiltration and immune evasion mechanisms [75]. Significant differences were defined in CD3+ and PD-1+ cell counts in epithelial contours between hereditary and sporadic MSI CRCs [76,77], as well as in CD8+ cells in epithelial contours of adenomas [78]. Severe infiltration, but without statistically significant difference, was observed in 20% of MSI-H patients and 12.8% of MSI-L/MSS patients. In contrast, severe infiltration of intra-tumor TILs was observed in 41.7% of MSI-H CRC patients and 4.3% of MSI-L/MSS patients, again indicating a close correlation between the extent of intra-tumor infiltration and MSI-H (*p* < 0.001) [79].

Regarding immune evasion mechanisms, HLA-I mediated antigen presentation occurs in approximately 70% of MSI CRCs [80,81,82]. This mechanism is abrogated by mutations in the β-2 globulin (*B2M*) gene found in 30% of MSI CRCs [83] and is strictly associated with an active immune response [84,85].

Other findings on tumor cells lacking HLA class II antigen expression due to mutations of regulatory genes such as Regulatory Factor X5 (*RFX5*), Class II Major Histocompatibility Complex Transactivator (*CIITA*), and Regulatory Factor X Associated Protein (*RFXAP*), seem to be favored in an environment of dense CD4+ T cell infiltration [86].

This evidence demonstrates that it is not sufficient to know the MS status of CRCs to predict the response to immune therapy, but close surveillance concerning the status of the immune phenotype of the same is necessary (Figure 1).

Legend: PD-1: programmed death-1; PD-L1: programmed death-1 ligand; CTLA-4: cytotoxic T-lymphocyte-associated protein-4; TCR: T-cell receptor; MHC: major histocompatibility complex; MMR: DNA mismatch repair. POLE: Polymerase Epsilon-DNA polymerase enzyme complex. Created by https://www.biorender.com/ (accessed on 2 February 2023).

## 3. Prognostic and Predictive Value of MMR defects

### 3.1. Direct Prognostic Implications

Pathological tumor staging remains the cornerstone for determining the prognosis and the (surgical and medical) treatments of CRC. Nevertheless, in recent years it has become evident that CRC has significant genetic and molecular heterogeneity even within the same stage. In this scenario, MSI is currently the most relevant molecular marker impacting prognosis (Table 1).

MSI as a favorable prognostic factor for CRCs translates into lower metastatic potential and thus, better survival [87]. This is exemplified by the different prevalence of MSI CRCs in non-metastatic stages (i.e., II and III; ≈20%) as compared with metastatic stages (stage IV; ≈2–5%) [88,89].

As confirmed by meta-analysis, MSI confers a survival advantage in terms of overall survival in real-world cohorts as well as in clinical trial populations, mainly in patients with locally advanced CRC (combined HR 0.65 [95% CI, 0.59 to 0.71]) with no lymph node involvement [88].

In stage II MSI CRC, both disease-free survival and overall survival are improved, especially in patients undergoing surgery alone [90], and pointing to this behavior was crucial to progressively modify the review of the adjuvant treatment of MSI CRC [91]. Conversely, in stage III MSI CRC, the prognostic value of MSI is not equally defined, and contradictory data have been reported in the literature [92,93].

The favorable effect of MSI on the prognosis seems progressively attenuated with the advancing tumor stage. Indeed, as concerns metastatic CRC (mCRC), some studies described a positive prognostic effect only in patients presenting a wild-type profile of BRAF [94], while other studies reported a better prognosis regardless of BRAF status [95].

For rectal cancer, the impact of MS status still needs to be clarified, and preliminary data have reported MSI as a less decisive prognostic factor [96]. Finally, in stage IV CRCs with dMMR, no prognostic advantage was found [97].

**Table 1 genes-14-01169-t001:** Report of the prognostic value of MSI in CRC. HR: Hazard Ratio; PFS: progression-free survival; OS: Overall survival. Data of PFS and OS are reported as median of years.

References	MSS			MSI		
	HR (95% C.I)	PFS (95% C.I)	OS (95% C.I)	HR (95% C.I)	PFS (95% C.I)	OS (95% C.I)
Malesci et al. [87]	NA	NA	NA	0.30 (0.16–0.54), *p* < 0.001, for DSS	NA	NA
Popat et al. [88]	NA	NA	NA	0.65 (0.59–0.71), pooled, for OS 0.67 (0.53–0.83), pooled, for PFS	NA	NA
Sankila et al. [89]	NA	NA	NA	0.64 (0.56–0.72), for OS in Lynch patients	NA	NA
Klingbiel et al. [92]	0.16 (0.04–0.64), *p* = 0.01, for OS in stage II	NA	NA	0.26 (0.10–0.65), *p* = 0.004, for OS in stage II	NA	NA
Venderbosch et al. [94]	1.34 (1.10–1.64), for PFS 1.94 (1.57–2.40), for OS	7.6 (7.3–8.0)	16.8 (16.3–17.5)	1.33 (1.12–1.57), for PFS 1.35 (1.13–1.61), for OS	6.2 (5.9–7.0)	13.6 (12.4–15.6)
Taieb et al. [95]	NA	NA	NA	0.82 (0.69–0.98), *p* = 0.029, for OS after recurrence	2.2 (1.9–2.7)	NA

### 3.2. Predictive Implications for Standard Adjuvant Treatment

As concerns clinical management, MS status has been demonstrated to predict the response or the resistance to classical adjuvant chemotherapy in stage II CRCs [94,98,99,100,101]. As proven by extracting data from phase-3 randomized clinical trials of FU-based therapy versus surgery-only controls, treatment benefit significantly diverged by MSI status, showing inferior outcomes for patients with MSI CRC who were treated with chemotherapy, as compared with those receiving surgery alone (*p* = 0.01) [102]. This evidence was endorsed in the large randomized phase III Quick and Simple and Reliable (QUASAR) trial, where around 2.200 patients with stage II CRC were randomized to receive adjuvant chemotherapy (5-FU based regimen) or observation only: the sub-analysis by MS status reported no benefit from adjuvant chemotherapy, and a significantly reduced risk of recurrence for MSI CRC as compared with MSS-CRC was shown (RR 0.53, 95%CI, 0.40–0.70; *p* < 0.001) [103,104]. Numerous meta-analyses validated the predictive value of MSI status both for therapy response and for relapse risk for stage II CRC [105,106]. Based on this evidence, patients with stage IIA MSI CRC are currently spared chemotherapy-related toxicities and reduced quality of life during treatment.

Diverging data on the predictive role of MSI in stage III CRC have been presented over the years, suggesting a likely shift in the clinical impact of MSI across disease stages. As fluoropyrimidine chemotherapy has been shown detrimental in stage II MSI CRC, the superiority in terms of response and overall survival of adding oxaliplatin to fluoropyrimidine regimens in patients with stage III MSI CRC remains controversial [107,108]. In detail, MSI was confirmed to be a positive predictive factor for patients in the subgroup N1 of stage III, while its prediction was lost in the N2 subgroup [109]. These data would suggest a variable effect from oxaliplatin-based adjuvant treatment in MSI stage III CRC patients.

Thus far, based on the available studies, the exclusion of patients with stage III MSI CRC from adjuvant treatment does not appear justified.

## 4. Actionability of Immune Responses as a Therapeutic Target in CRC according to MS Status

### 4.1. Immune Checkpoint Targeting

Recent strategies exploiting immunotherapy are based on restoring the functional activation of effector T-cells for the revitalization of an efficient adaptive immune response. Acting as crucial modulator of the immune response across different cancer types, IC molecules [110] are the main target of the attempt to rescue an immune control of neoplastic growth.

Programmed-death 1 (PD-1, also known as CD279) is a surface receptor of CD8+ and CD4+ T cells that belongs to the immunoglobulin superfamily and by preventing cell insults that cause irreversible damages [111] plays an important role in constraining unwarranted inflammatory reactions [111,112,113]. PD-1 signaling is activated by its two ligands PD-L1 and PD-L2 [114,115], which can be expressed in myeloid cells as well on tumor cell surface, thus representing a mechanism by which they can steer the microenvironment toward an immunosuppressive one [113,116]. High levels of PD-1 expression on CD8+ T cells are linked to the loss of effector functions, including the ability of to proliferate and release IL-2, TNF-α, and INF-γ, thus restraining the ability of TILs to carry out their antitumor immune responses [117]. Initially, it was shown that the clinical activity of the monoclonal antibody pembrolizumab, by blocking programmed cell death 1 (PD1) on T cells and other immune cell populations, enhanced the cytotoxic killing of tumor cells, improving survival rates in metastatic MSI CRCs [118,119,120].

The use of anti-PD1 antibodies in CRC was initially discouraged, especially after initial studies in PD-1-deficient mice suggesting that PD-1 deficiency increases the incidence of autoimmune diseases [121,122], despite small benefits being obtained. Furthermore, in the phase I study by Brahmer et al., only 1 in 14 patients had a complete response [123], but noticeably that only responding patient had a MSI CRC [124].

Only in 2017, the FDA granted accelerated approval for the first two PD-1 monoclonal antibodies (moAbs) to treat MSI-H or dMMR CRC: Opdivo, also known as nivolumab, MDX-1106, BMS-936558, and ONO-4538 (Bristol-Myers Squibb) and Keytruda, also known as pembrolizumab, lambrolizumab, and MK-3475 (Merck) [125] (Figure 2).

Regarding pMMR and dMMR metastatic CRC, despite clinical trials studying different combinations [118], the knowledge of the extent of cooperative interaction between checkpoints remains limited.

Ma and coll. found that the expression levels of 43 IC genes strongly correlated with each other, serving as the theoretical basis for the combination of checkpoint blockade. Investigating the relationship between IC genes and other biomarkers of immunotherapy responsiveness, they also found a significant association between the expression of a series of IC genes and CRC prognosis, as well as between biomarkers of responsiveness, neoantigens, and MMR status. For example, an important member of the B7 transmembrane protein family (B7-H4), the Indoleamine 2,3-Dioxygenase 1 (IDO1) coding gene, the Lymphocyte Activating 3 (LAG3), the Cluster of Differentiation 48 antigen (CD48), and TNF Receptor Superfamily Member 9 (TNFRSF9) had a significant positive correlation with dMMR, a high tumor mutational burden. These findings give rise to the possibility of stratifying CRC based upon the probability of responsiveness to IC inhibitors (ICIs) [133].

Cytotoxic T-Lymphocyte Antigen 4 (CTL4A—also known as CD152) is a receptor belonging to the Ig superfamily expressed on recently activated CD4+ and CD8+ T lymphocytes. Following the binding to one of its ligands, B7 (or L2)-1 (also known as CD80) or B7-2 (or CD86), expressed on professional antigen-presenting cells (APCs), CTLA-4 transmits an inhibitory signal within the lymphocyte, contributing to the homeostatic regulation of the immune response [111,134]. As seen in pre-clinical study, CTLA-4 engagement inhibits IL-2 accumulation and cell cycle progression upon activation of resting T cells, in the absence of the induction of apoptotic cell death [116,135].

Other studies have revealed that the development of CRC is related to the epigenetic silencing and regulation of ICs impacting antigen processing and presentation by tumor cells and facilitating their evasion [136]. An example is the combination effect of PDCD-1 and LAG-3 IC genes, which may potentially serve as blood-based predictive biomarkers for CRC risk [137].

### 4.2. An Evolving Therapeutic Scenario

The fact that MSI CRCs have a peculiar immunological microenvironment, with activated CD8(+) cytotoxic T lymphocyte (CTL) as well as activated Th1 cells characterized by IFNγ production, was recapitulated by Llosa et al. [32], who also clearly pointed out that MSI CRCs were also marked by the upregulation of numerous ICs (i.e., PD-1, PD-L1, CTLA-4, LAG-3, and IDO), all targetable by selective inhibitors [32].

The descending clinical hypothesis that tumors with high rates of somatic mutations due to MMR defects may be susceptible to IC blockade was eventually tested in a phase 2 trial comparing the responsiveness to the treatment with the anti-PD-1 ICI pembrolizumab of progressive metastatic CRCs (and other carcinoma), with or without MMR defects. NGS analysis documented a mean of 1782 somatic mutations/tumor in MMRd cancers as compared with 73/tumor in MMRp ones [128].

The median progression-free survival and median overall survival were not reached in the patients with MSI CRCs; however, the comparison of the cohorts with MSI and MSS CRCs showed hazard ratios for disease progression or death (0.10; 95% CI, 0.03 to 0.37; *p* < 0.001) that favored patients with MSI CRCs [128]. Patients with MSI sporadic cancers had a higher rate of objective response than those with cancers occurring in the context of LS.

Response rates of 30–40% MSI for single-agent anti-PD-1 antibodies (pembrolizumab or nivolumab) and of 40–50% for the combination of nivolumab and the anti-CTLA-4 antibody (i.e., ipilimumab) in MSI CRCs has been demonstrated also in later lines of treatment [118,119]. The cumulative evidence on efficacy and durability led to the approval as salvage therapy in MSI CRCs of pembrolizumab and nivolumab (with or without ipilimumab) in 2017 and 2018, respectively.

Lately, the paradigm of immunotherapy moved toward the idea of a potential curative approach with ICI in advanced MSI CRCs, establishing pembrolizumab as first-line therapy and becoming a prominent biomarker-based treatment in gastrointestinal oncology. Immunotherapy also has the potential of becoming the upfront treatment in operable MSI CRCs [129] and in MSI rectal cancers cohorts [138,139].

## 5. Immunotherapy and MSI Colorectal Cancer: On-Stage and Incoming Broadcasting

The use of anti-PD1 mAb has been tested alone or in association with the classic chemotherapy regimens for the clinical management of patients with CRC (Figure 3).

The effectiveness of adjuvant chemotherapies that inhibit the proliferation of cancer cells, such as OXA and 5-FU, can also produce immunomodulatory effects, such as immune-mediated cell death. While 5-FU can facilitate antigen uptake by dendritic cells (DCs) and selectively kill monocyte-derived suppressor cells (MDSCs), sparing other lymphocyte subtypes [148,149], OXA increases PD-L1 expression on tumors and DCs, inducing immunogenic cell death with the release of damage-associated molecular patterns) [150]. Interestingly, tumor-associated macrophages have been suggested as a potential biomarker of responsiveness to neoadjuvant therapy with 5-FU alone or in combination with OXA in stage III CRC [151,152].

To quantify the effect of the changes induced by chemotherapy in the immune environment and how they affect ICIs, non-invasive molecular imaging with radio-labelled granzyme B target peptides (GZBs) is the best method to stratify the response to ICIs administered alone or in combination [153,154,155].

GZB target peptides may stratify the response to chemotherapies that exert an immune-stimulating effect alone or in combination with ICI. It has been seen that tumors sensitive to chemo-ICI combinations PD1 + OXA or PD1 + 5-FU showed even greater absorption than those treated with monotherapy alone. While the first had a high increase in CD8+ and CD8 + GZB+, the latter had a significant increase in tumor-infiltrating NK+ and GZB + NK+ cells [153].

Recently, the predictive value of MSI for immunotherapy in non-metastatic and mCRC has been also proved. The phase-2 CheckMate 142 study confirmed the long-term benefit of Nivolumab (anti-PD-1) plus low-dose Ipilimumab (monoclonal antibodies that target CTLA-4) for previously treated patients with MSI-mCRC [141,156] and the phase-3 KEYNOTE-177 trial assessed that Pembrolizumab (anti-PD-1) led to a significantly more durable progression-free survival than chemotherapy when received as first-line therapy for MSI-mCRC [128,140]. Pembrolizumab is currently recommended as first choice treatment for patients with MSI-mCRC, and Ipilimumab plus Nivolumab regimen is recommended for MSI-mCRC patients progressing after first-line treatment [157]. Regarding earlier stage CRCs, in the NICHE study investigating Nivolumab plus Ipilimumab with neoadjuvant intent followed by surgery, 95% of the patients (19/20) showed major pathological response (defined as ≤10% residual viable tumor in the surgical specimen), and among them a pathological complete response was observed in 12/20 (60%) [129]. Data from a prospective phase II study investigating the neoadjuvant immunotherapy with Dostarlimab (an anti-PD-1 monoclonal antibody) of locally advanced (stage II or III) rectal adenocarcinoma reported that among 16 enrolled patients, 12 completed treatment, and all these (100%) presented clinical complete response defined as no evidence of residual tumor at imaging, endoscopy, or biopsy [132]. Notably, at the time of the published follow-up (range, 6–25 months), no progression or recurrence were observed, and no patients had received chemo-radiotherapy or undergone surgery [132]. Another noncomparative, randomized phase II trial (PICC) evaluated the efficacy of the anti-PD-1 Toripalimab in association or not with Celecoxib (a selective non-steroidal anti-inflammatory drug) for patients affected by locally advanced MMRd/MSI-high CRC. Although a long follow-up would be necessary to assess effects on survival-related endpoints, its findings suggest that this different treatment was associated with a high rate of pathological complete response and an acceptable safety profile, without compromising surgery [131].

Other trials aim to target cytokines and chemokines in combination with ICIs. A clinical study on patients with metastatic pMMR CRCs aimed to combine ICI with a synthetic complex of carboxymethylcellulose, polyinosinic-polycytidylic acid, and poly-L-lysine double-stranded RNA (poly-ICLC), a molecule that stimulates cytokine release by inducing INF-α and -γ production. In this clinical trial, the investigators hypothesize that treating patients having pMMR CRC with immune-stimulating agent poly-ICLC will generate an inflammatory response, increasing epitope recognition and the development of tumor-reactive T-cells at the tumor site. However, INF-α and -γ produced by the poly-ICLC will increase PD-L1 expression and limit new T-cell development, suggesting that, PD1 blockade could increase the effectiveness of treatment with Pembrolizumab [158].

Pembrolizumab is also associated with Olaptesed pegol (NOX-A12), which targets the C-X-C motif chemokine 12 (CXCL12), also known as stromal cell-derived factor 1 (SDF-1), which activates leukocytes and are often induced by proinflammatory stimuli such as lipopolysaccharide, TNF, or IL1. In cancer, CXCL12 acts as a communication bridge between tumor cells and their environment, conferring resistance to ICI through T-cell exclusion in preclinical models. The hypothesis is that inactivation of CXCL12 by NOX-A12 induces changes in the tumor microenvironment of patients with CRC and pancreatic cancer, which would render tumors more susceptible to checkpoint inhibition [159].

To enhance immunotherapy in CRC characterized by low immunogenic infiltration, complementary therapeutic approaches are needed. An example is the clinical trial reported on pMMR/MSS CRC treated with Cibisatamab, a T-cell antibody (TCB) that simultaneously binds CEA on tumor cells and CD3 on T-cells. Evidence supported an enhanced antitumor activity in advanced CRC and other CEA-expressing tumors in combination with Atezolizumab [160,161]. Another study targets LAG3 for the first time in a phase I trial [162] on advanced MSS-CRC with anti-LAG-3 antibody MK4280 (Favezelimab) plus Pembrolizumab with promising antitumor activity, differing from the single therapy [162].

Cell therapies based on chimeric antigens receptors (CARs) expressed by T-cells are being tested on solid tumors and eventually could represent an encouraging modality of CRC clinical management [163].

## 6. Conclusions

In this review, we elucidated the present evidence and the evolved knowledge on the prognostic and predictive value of MSI in CRCs that has led to the very first and unbeaten example of personalized medicine in oncology. Altogether, data support MSI as the leading molecular marker with clinical value in early-stage CRC; no further comparable molecular stigma has been incorporated into the management algorithms of CRC yet.

Despite the heterogeneity of the data, one observation appears consistent, which is that the prognostic impact of dMMR/MSI declines with regional and distant metastatic disease is such that a favorable prognosis exists in stage II CRC, while the effects diminish in stage III disease.

At the same time, in advanced disease, MSI has moved forward, becoming the predictive marker for responsiveness to immunotherapy targeting IC blockade, thus opening new scenarios aside from standard adjuvant and neo-adjuvant treatments for pMMR CRC.

After having had the chance to learn so many lessons from MSI CRC, one may ask what could be expected ahead. On one hand, we should be eager to see whether neo-adjuvant immunotherapy will gain major attention as first-line option for MSI CRC. This would have great impact on tumor test assessing MSI/MMR defects at time of diagnosis rather than after resection, thus requiring great changes in diagnostic rather than therapeutic pipelines. On the other hand, basic science should continue to explore whether the therapeutic manipulation of CRC could be exploited to induce MSI [164], then allowing the employment of immunotherapy as an additional strategy for the therapy of CRC at large.

## Figures and Tables

**Figure 1 genes-14-01169-f001:**
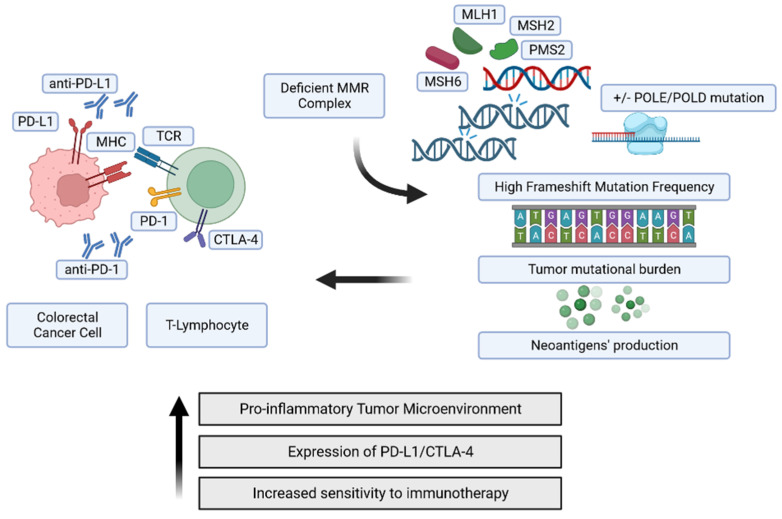
Molecular pathway of DNA mismatch repair deficient tumors. Figure 1 shows the molecular effects of a defective mismatch repair system that accumulates DNA frameshift mutations, producing and releasing soluble neoantigens (peptides and proteins). Consequently, the immune system is over-stimulated, a shift toward a pro-inflammatory tumoral microenvironment is observed, and the sensitivity of immunotherapy increases.

**Figure 2 genes-14-01169-f002:**
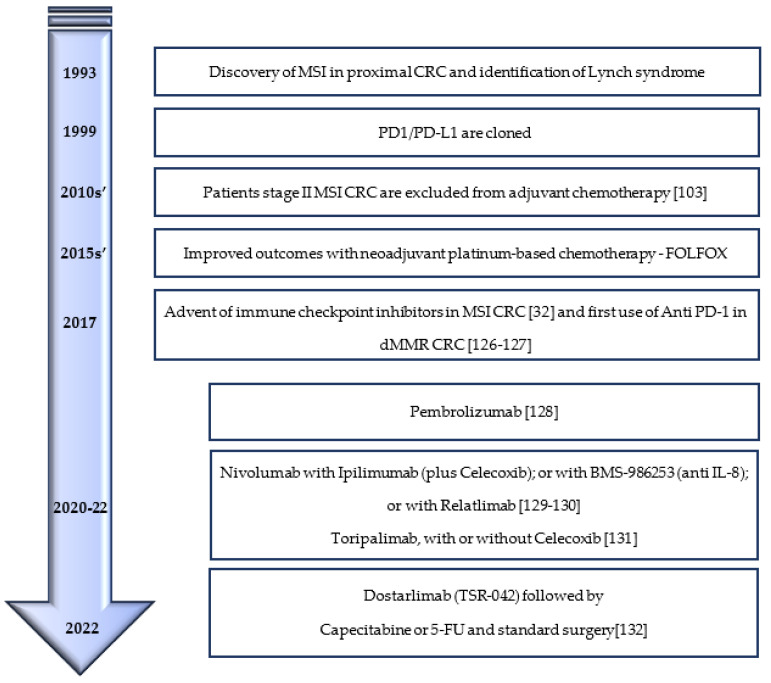
Timeline of therapeutical advances in MSI CRC [32,103,126,127,128,129,130,131,132].

**Figure 3 genes-14-01169-f003:**
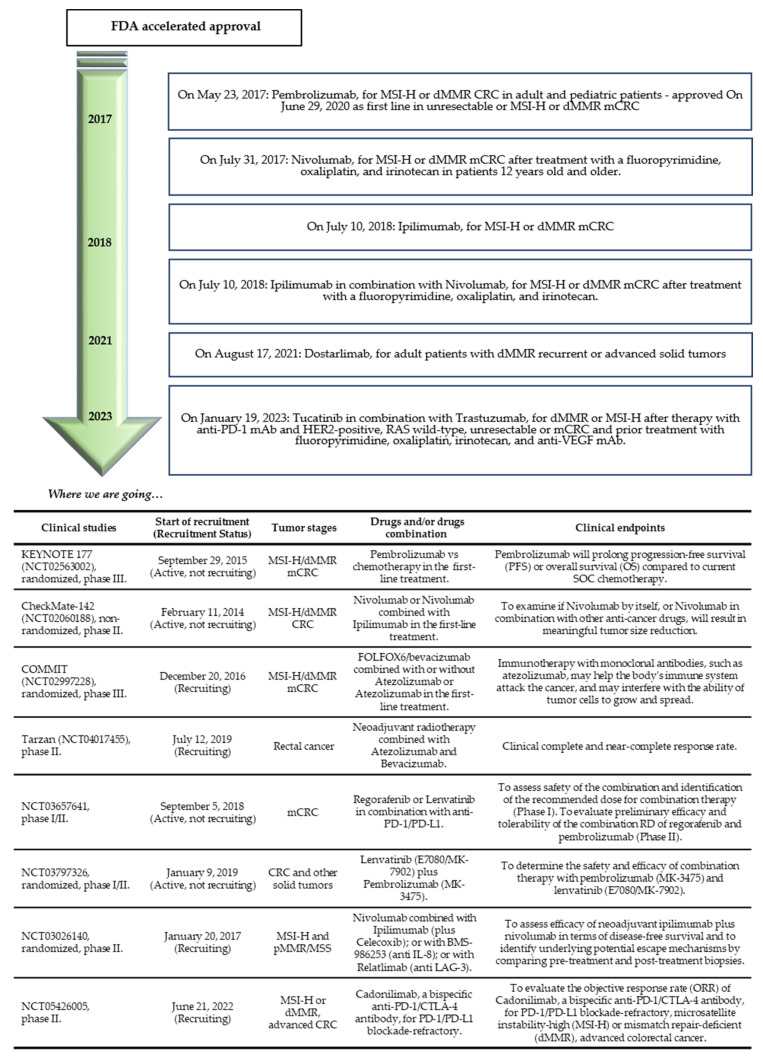
Upper panel: timeline of FDA accelerated approval; lower panel: table of ongoing clinical trials [118,128,129,130,140,141,142,143,144,145,146,147].

## Data Availability

Not applicable.

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
