# Peer review of "Microsatellite Instability and Immune Response: From Microenvironment Features to Therapeutic Actionability—Lessons from Colorectal Cancer"

_genes, 2023, doi:10.3390/genes14061169_

Round 1

Reviewer 1 Report

The review "Microsatellite instability and immune response: from microenvironment's features to the therapeutic actionability" by Luana Greco and co-authors is well-written and provides a good summary of the current state of knowledge in the field of colon cancer biology. It also describes connotations between MSI status and treatment of colon cancer which might be of use for healthcare professionals and research scientists. I do not find any factual flaws that might disqualify publication of this manuscript. 

During the review I found a small typographical error in the Figure 1 ("T-lymphocite" instead of "T-lymphocyte") which should be corrected during the editorial process.

Author Response

We thank the reviewer for his positive comment and for the note reported. We have corrected all the typos.

Reviewer 2 Report

Specific comments to the authors

The submitted review “Microsatellite instability and immune response: from microenvironment’s features to therapeutic actionability” gathers, summarize and analyses heterogeneous aspects of the interaction of microsatellite instability and immune response in relation to biomarker value and possibly therapeutic options based on already published reviews in-vitro and in-vivo experiments as well as clinical trials.

The presented topics range from classical clinico-pathological findings/characteristics and treatment concepts in cases with microsatellite instability now and in the future. In summary the author gives an interesting survey of microsatellite instability and immune related therapy escpecially in colorectal cancer, which is mostly easy to read, to follow and to understand. The authors should clarify some aspects before accepting the manuscript for publication as mentioned below.

 Overall

The numeration of the chapters are not correct: 5. Immunotherapy and MSI colorectal cancer: on-stage and incoming broadcasting” and “5. Conclusions“. Please correct.

# Title: As the aspects of microsatellite instability and immune response are mostly linked to cases with colorectal cancer, the title should be complemented like “lessons from the colorectal cancer”.

# Abstract: The authors should emphasise that the interaction between microsatellite instability and the immune response is a fundamental mechanism in some human tumurs, which opens up a very large opportunity for specific immune-based therapy. What is the meaning of “PD-1 e PD-L1” (also found throughout the manuscript) or just a typo?

# Introduction: The authors stated, that they “aim to summarize the peculiar features of MSI tumors”, whereby they focused on colorectal cancer. Please specify.

# Chapter “2.1 MSI, a distinguishable genetic attractor of the adaptive immune response.”: Please add a table with all mentioned immunological markers and associated immune linked cell type in colorectal cancer cases with MSI in relation to clinico-pathological findings. Regarding Figure 1 correct the typo “Lymphocite”. The reading directions of the figure 1 is not clear. Please add arrows.

# Chapter “3.1. Direct prognostic implications”: This chapter is largely superficies to demonstrate the real prognostic potency of MSI. Therefore, please add a table containing hazard ratios with confidence interval as well as median progression-free survival and median overall survival of cases with or without MSI in dependency of tumor stage.

# Chapter “5. Immunotherapy and MSI colorectal cancer: on-stage and incoming broadcasting” Please summarize the mentioned clinical studies with more details (not only descriptive as included in figure 3): beginning of recruitment, drugs and drugs combination, clinical endpoints, tumor stages.

# Conclusion: The authors should add a statement of milestones to be achieved in the next five years to improve our understanding of microsatellite instability and the immune response, particularly in cases of colorectal cancer.

Moderate editing of English language

Author Response

We thank the reviewer for his positive comment and for the note reported. We have corrected all the typos and we made changes to the text and visible from the font in red.

Overall

The numeration of the chapters are not correct: 5. Immunotherapy and MSI colorectal cancer: on-stage and incoming broadcasting” and “5. Conclusions“. Please correct.

Answer: We thank the reviewer for the note. We have corrected the numbering.

# Title: As the aspects of microsatellite instability and immune response are mostly linked to cases with colorectal cancer, the title should be complemented like “lessons from the colorectal cancer”.

Answer: We thank the reviewer for the suggestion. We updated the title with "Lessons from the colorectal cancer."

# Abstract: The authors should emphasise that the interaction between microsatellite instability and the immune response is a fundamental mechanism in some human tumurs, which opens up a very large opportunity for specific immune-based therapy. What is the meaning of “PD-1 e PD-L1” (also found throughout the manuscript) or just a typo?

Answer: We thank the reviewer for the note. We have defined in full the meaning of PD-1 and PD-L1 which are not a typo.

# Introduction: The authors stated, that they “aim to summarize the peculiar features of MSI tumors”, whereby they focused on colorectal cancer. Please specify.

Answer: We thank the reviewer for the suggestion. We have specified our focus.

# Chapter “2.1 MSI, a distinguishable genetic attractor of the adaptive immune response.”: Please add a table with all mentioned immunological markers and associated immune linked cell type in colorectal cancer cases with MSI in relation to clinico-pathological findings. Regarding Figure 1 correct the typo “Lymphocite”. The reading directions of the figure 1 is not clear. Please add arrows.

Answer: We thank the reviewer for the suggestion. We believe that inserting a table would only be redundant information as all mentioned immunological markers are well defined in the text. About Figure 1, we added arrows to improve reading and corrected typos.

# Chapter “3.1. Direct prognostic implications”: This chapter is largely superficies to demonstrate the real prognostic potency of MSI. Therefore, please add a table containing hazard ratios with confidence interval as well as median progression-free survival and median overall survival of cases with or without MSI in dependency of tumor stage.

Answer: We thank the reviewer for the suggestion. We have added a table with the required information. Unfortunately, regarding MS-stable cases, information was not available for all the studies mentioned.

# Chapter “5. Immunotherapy and MSI colorectal cancer: on-stage and incoming broadcasting” Please summarize the mentioned clinical studies with more details (not only descriptive as included in figure 3): beginning of recruitment, drugs and drugs combination, clinical endpoints, tumor stages.

Answer: We thank the reviewer for the suggestion. We have reported information from ongoing trials as indicated. Figure 3 has therefore been modified by adding a table from the bottom panel of the same.

# Conclusion: The authors should add a statement of milestones to be achieved in the next five years to improve our understanding of microsatellite instability and the immune response, particularly in cases of colorectal cancer.

Answer: We thank the reviewer for the note. We have added a statement of milestones as suggested.